# *Salmonella* spp. in Pet Reptiles in Portugal: Prevalence and Chlorhexidine Gluconate Antimicrobial Efficacy

**DOI:** 10.3390/antibiotics10030324

**Published:** 2021-03-19

**Authors:** João B. Cota, Ana C. Carvalho, Inês Dias, Ana Reisinho, Fernando Bernardo, Manuela Oliveira

**Affiliations:** CIISA–Centro de Investigação Interdisciplinar em Sanidade Animal, Faculdade de Medicina Veterinária, Universidade de Lisboa, Av. da Universidade Técnica, 1300-477 Lisboa, Portugal; accarvalho@fmv.ulisboa.pt (A.C.C.); ines.adpdias@gmail.com (I.D.); anareisinho@fmv.ulisboa.pt (A.R.); fbernardo@fmv.ulisboa.pt (F.B.); moliveira@fmv.ulisboa.pt (M.O.)

**Keywords:** *Salmonella*, reptiles, isolation, antimicrobial resistance, biofilms, chlorhexidine gluconate, public health

## Abstract

A fraction of human *Salmonella* infections is associated with direct contact with reptiles, yet the number of reptile-associated Salmonellosis cases are believed to be underestimated. Existing data on *Salmonella* spp. transmission by reptiles in Portugal is extremely scarce. The aim of the present work was to evaluate the prevalence of *Salmonella* spp. in pet reptiles (snakes, turtles, and lizards), as well as evaluate the isolates’ antimicrobial resistance and virulence profiles, including their ability to form biofilm in the air-liquid interface. Additionally, the antimicrobial effect of chlorhexidine gluconate on the isolates was tested. *Salmonella* was isolated in 41% of the animals sampled and isolates revealed low levels of antimicrobial resistance. Hemolytic and lypolytic phenotypes were detected in all isolates. The majority (90.63%) of the *Salmonella* isolates were positive for the formation of pellicle in the air-liquid interface. Results indicate chlorhexidine gluconate is an effective antimicrobial agent, against the isolates in both their planktonic and biofilm forms, demonstrating a bactericidal effect in 84.37% of the *Salmonella* isolates. This study highlights the possible role of pet reptiles in the transmission of non-typhoidal *Salmonella* to humans, a serious and increasingly relevant route of exposure in the *Salmonella* public health framework.

## 1. Introduction

*Salmonella* is a well-known food-borne illness etiological agent, reported as the second most common zoonotic agent, causing 91,857 confirmed cases of disease in the European Union during 2018 [1] and an estimated number of 93.8 million cases worldwide annually [2]. The clinical manifestations of human salmonellosis are frequently those associated with a self-limited gastroenteritis, namely nausea, vomiting and diarrhea, but can also include severe complications, including bacteremia and extra-intestinal infections [3]. Though most commonly associated with contaminated food, human salmonellosis can also occur through the contact with infected animals, such as farm animals and pets, including reptiles [4]. 

In the course of the past years, reptiles have been increasingly regarded as household pets, with their estimated numbers ascending up to 8 million only in the Europe Union in 2019 [5]. *Salmonella* not only can be found in the gastrointestinal tract of healthy reptiles, but also in the environments where those animals are kept [6,7]. *Salmonella enterica* subspecies *enterica* is commonly found in warm-blooded animals, while the remainder subspecies, *salamae*, *arizonae*, *diarizonae*, *houtenae*, and *indica*, along with *Salmonella bongori* are frequently isolated either from reptiles or from the environment [8]. Furthermore, among more than 2500 known *Salmonella* serotypes, over 40% are associated with reptiles and are rarely isolated from other animals, including humans [9]. Although infrequent when compared with food-borne cases, accounting for 6% of all human salmonellosis cases both in the USA and in Europe [10], reptile-associated salmonellosis (RAS) seems to be more related with more severe clinical scenarios, such as systemic and severe disease development, especially in children, elderly people, and pregnant women [6]. In fact, RAS is a growing public health concern worldwide, with different reports pointing out for its role in disease outbreaks [11,12]. Despite the several RAS cases that have been reported in different European countries [13], there seems to be no available data regarding Portugal. 

As observed for non-typhoidal salmonellae of other sources, there has been an increasing focus on antimicrobial resistance in reptile-associated *Salmonella* [14,15,16] since this feature can impair the success of treatments of both human and veterinary *Salmonella* infections [17]. Antimicrobial resistance can either arise from mutations in chromosomal genes (intrinsic resistance), which are caused by selective pressure, or through the acquisition of antimicrobial resistance determinants encoded in plasmids (extrinsic resistance), by horizontal transfer [18]. The role of reptiles as disseminators of antimicrobial resistant (AMR) *Salmonella* has been suggested [19,20]. Furthermore, *Salmonella* is known to have the ability of producing biofilms in different biotic and abiotic surfaces [21]. Not only are bacterial cells in biofilms more tolerant to antimicrobials when compared with the corresponding planktonic cells [22] but also more resistant to several chemical disinfectants [21]. 

Chlorhexidine is a biocide widely included in antiseptic products, especially in handwashing and oral products, due to its broad-spectrum efficacy and low irritability [23]. For surgical skin preparations and hand scrub, chlorhexidine is available in 4% solutions, while for wound cleaning is used as a 0.5% concentrated solution [24]. In veterinary care, chlorhexidine gluconate is a common disinfectant. In reptile treatment, chlorhexidine solutions are frequently used for topical application and preoperative scrubs, in concentrations below 2% [25], but there is a lack of clear guidelines regarding the most appropriate concentration to use.

The aim of the present study was to assess the presence of *Salmonella* spp. among the intestinal microbiota of pet reptiles in the Metropolitan area of Lisbon, Portugal, and to characterize those isolates, regarding antimicrobial susceptibility and virulence traits, bringing more information on the role of reptile-associated *Salmonella* on the public health scenario. Additionally, the antimicrobial efficacy of chlorhexidine gluconate against both planktonic cells and biofilms was also evaluated.

## 2. Results

### 2.1. Salmonella spp. Isolates

Of the 78 reptiles sampled 32 were identified as *Salmonella* positive (41%), specifically four Ophidians (50%), 14 Saurians (51.9%), and nine Chelonians (20.9%), belonging to 12 different owners (Table 1). Overall, the *Salmonella* recovery rates where higher both in Ophidians and Saurians when comparing with the one recorded in Chelonians (*p* = 0.016). After assessing the biochemical profile using API20E strip tests of the presumptive *Salmonella* isolates, 13 were identified as *Salmonella enterica* subspecies *arizonae* and 19 as *Salmonella* spp. (Table 2). 

More than half of all *Salmonella* positive animals (62.5%) were detained by only three owners (E, F, and J). Moreover, owner J alone kept 12 *Salmonella* positive reptiles, more specifically Saurians. *Salmonella* isolates from co-habiting animals belonged to similar species with the exception for the isolates recovered from the animals of owner J, where the majority was identified as *Salmonella enterica* subspecies *arizonae* (10/12) and the remaining as *Salmonella* spp. (2/12) (Table 2). Notably, whenever an owner possessed multiple *Salmonella* positive animals, those animals belonged to the same reptile group.

### 2.2. Antimicrobial Resistance

All of the studied isolates were susceptible to gentamicin (CN) and ciprofloxacin (CIP) (Table 3). High levels of susceptibility to amikacin (AK) (96.87%), sulfamethoxazole/trimethoprim (SXT) (96.87%), nalidixic acid (NA) (93.75%), enrofloxacin (ENR) (90.63%), amoxicillin/clavulanic acid (AMC) (90.63%), ampicillin (AMP) (90.63%), cefotaxime (CTX) (87.50%), tetracycline (TE) (87.50%), and to chloramphenicol (C) (81.25%) were also recorded. On the other hand, 31 of the *Salmonella* isolates (96.87%) were resistant to penicillin (P).

When comparing groups, resistance to AMC (*p* = 0.0286) and AMP (*p* = 0.0286) were associated with Chelonian *Salmonella* spp. isolates, as resistance to both antimicrobials was only detected, and simultaneously, in isolates 26, 36, and 47, all originating from turtles of different owners (Appendix A). No other statistically significant differences regarding antimicrobial susceptibility were detected. 

Only three isolates (9.37%), all from Chelonians, were resistant to three or more of the antimicrobial compounds tested (isolates 26, 36, and 47) (Appendix A). The multiple resistance patterns were AMC/AMP/P, observed in isolates 26 and 36, and AMC/AMP/P/TE, revealed by isolate 47. None of the isolates was considered to be multidrug resistant, since the detected resistance patterns included antibiotics from the same class.

### 2.3. Virulence Phenotype

Virulence phenotypic testing revealed that all of the isolates studied expressed both hemolytic and lipolytic behaviors (Table 3). Contrarily, gelatinase activity was not detected in any of the *Salmonella* isolates studied. Overall, DNase activity was observed in more than half (59.37%) of the isolates. No statistically significant differences in phenotypical behavior were identified when comparing isolates from different animal groups.

### 2.4. Minimum Inhibitory Concentration and Minimum Bactericidal Concentration

The minimum inhibitory concentration (MIC) and minimum bactericidal concentration (MBC) values of chlorhexidine gluconate calculated for each isolate can be found on Appendix A.

The overall average MIC value was 11.90 mg/L ± 3.68, ranging from 8.16 mg/L (MIC value observed towards a Chelonian isolate), to 23.81 mg/L (MIC value towards a Chelonian and a Saurian isolates, all from different owners), with a median value of 10.72 mg/L. The majority of the chlorhexidine gluconate MIC values (75%) calculated for each *Salmonella* isolate only ranged between 9.52 mg/L and 14.29 mg/L. When comparing groups, the average MIC values regarding Ophidian, Chelonian, and Saurian isolates were 11.98, 11.25, and 12.19 mg/L, respectively, the differences were not statistically significant (*p* = 0.802) (Table 4).

Regarding MBC, the overall mean value was 38.8 mg/L ± 50.25, with a minimum value of 9.52 mg/L (observed towards a Chelonian isolate), and a maximum value of 247.62 mg/L (regarding a Saurian isolate), with a median value of 23.22 mg/L. Although a high variability in MBC values was found, towards half of the studied isolates those values ranged between 11.91 mg/L and 23.81 mg/L. When comparing groups, the average MBC values obtained regarding the Ophidian isolates, 86.84 mg/L, the Chelonian isolates, 27.87 mg/L and the Saurian isolates, 33.87 mg/L, were not statistically different (*p* = 0.257). 

Chlorhexidine gluconate demonstrated to have a bactericidal effect in the majority of the *Salmonella* isolates (84.37%), since only five isolates (15.63%) had MBC/MIC ratio above 4 (Appendix A).

### 2.5. Biofilm Formation in the Air-Liquid Interface

The biofilm formation capability of the *Salmonella* isolates obtained from pet reptiles was studied by observing the development of a pellicle in the air-liquid interface. Of all isolates, only three (9.37%) were not able to form biofilms, thus the vast majority (90.63%) formed a clearly detectable biofilm. The shortest period required for biofilm formation was three days, and the longest was six days. The average number of days until the biofilm was formed was 4.4 days ± 0.90, and the majority of the isolates (75.9%) were able to form the biofilm in five days or less.

The differences on the average number of days until biofilm formation by Ophidian (5.1 days), Chelonian (4.7 days), and Saurian isolates (4.2 days) were considered not to have statistical significance (*p* = 0.211) (Table 4).

### 2.6. Minimum Biofilm Inhibitory Concentration and Minimum Biofilm Eradication Concentration Determination

The minimum biofilm inhibitory concentration (MBIC) and minimum biofilm eradication concentration (MBEC) values of chlorhexidine gluconate regarding each isolate can be found on Appendix A.

The MBIC values ranged from 14.29 mg/L to 232.15 mg/L, with an average value of 68.41 mg/L ± 32.68, and a median value of 71.43 mg/L. Despite the broad range of values, 71.43 mg/L of chlorhexidine gluconate was the MBIC value for more than half (59.4%) of the isolates tested. When comparing groups, the recorded average MBIC values regarding Ophidian, 57.15 mg/L, Chelonian, 64.02 mg/L, and Saurian isolates, 72.87 mg/L, did not statistically differ (*p* = 0.509) (Table 4).

Concerning the MBEC values, the average chlorhexidine gluconate biofilm eradication concentration was 360.08 mg/L ± 235.18, with a minimum of 33.34 mg/L and a maximum of 714.29 mg/L, and a median value of 392.86 mg/L. Regarding six isolates, one Ophidian, one Chelonian, and four Saurian related isolates, the MBEC values were considered to be greater than the highest concentration tested, therefore, the results were expressed as >714.29 mg/L. 

## 3. Discussion

Several research groups from multiple countries have reported the isolation of *Salmonella* spp. from pet or captive reptiles, including turtles, lizards, and snakes [26,27,28,29,30,31,32]. Although this is not a recent issue, to the author’s best knowledge, the present report is the first regarding the isolation of *Salmonella* spp. from healthy pet reptiles in Portugal. Our results point out to an overall *Salmonella* spp. prevalence of 41%, which is similar to studies performed with captive or pet reptiles in Australia (47%) [32], Spain (48%) [14], Norway (43%) [33], or Sweden (49%) [7], but higher than reports from smuggled reptiles in Taiwan (30.9%) [15] or captive animals in Croatia (13%) [29] or in New Zealand (11.4%) [31]. Furthermore, in our study, the prevalence of *Salmonella* spp. was higher in both Ophidians (50%) and Saurians (51.9%), when compared with Chelonians (20.9%) (*p* = 0.016). The lower isolation rates in turtles when compared with other reptiles can be associated with seasonal variations, observed when turtles are preparing for hibernation [28], but also with the diet of these animals [12,15,16]. In fact, the sample collection period occurred before the hibernation stage of Chelonians, during the colder months of the year. Nevertheless, the impact of pet turtles in the reptile-associated salmonellosis scenario should not be underestimated, since exposure to *Salmonella* positive turtles has been linked to disease outbreaks [34,35,36]. 

High levels of antimicrobial susceptibility to the majority of the antibiotics tested were found in most the *Salmonella* isolates, and only three isolates (9.37%) were resistant to three or more of the compounds tested. Our results differ from those reported in a recent study carried out in Spain, in which 72% of the isolates were considered to be multidrug resistant [14]. *Salmonella* isolates from reptiles are known to be resistant to several antibiotics frequently used in therapy. This not only implies that reptiles can shed multidrug resistant salmonellae to the environment and to other animals, including humans, but also the genes responsible for those antimicrobial resistances could be transferred to other enteric bacteria [17].

All the isolates studied expressed both hemolytic and lipolytic behaviors on plate tests. These two virulence phenotypes should be further investigated. Hemolysis is not associated with human non-typhoidal salmonellosis cases, and it has not been reported as a virulence trait by other authors, though it was shown that the hemolytic activity in *Salmonella enterica* serovar Typhimurium is dependent of the pathogenicity island 1 type III secretion system [37]. Extracellular lipases have been proposed as potential virulence factors in other pathogenic bacteria, such as *Staphylococcus aureus*, *Staphylococcus epidermis*, or *Pseudomonas aeruginosa* [38], though their role in *Salmonella* spp. virulence does not seem to be fully studied [39]. DNase testing pointed out the presence of extracellular desoxiribonucleases in more than half of the isolates. Gelatinase activity was not detected, even though it is a biochemical characteristic of *Salmonella enterica* subsp. *arizonae* [40]. It is possible that the analyzed isolates harbored the gene responsible for gelatin digestion, even though the isolates under the present study conditions did not express that phenotype. Recently, Salmonellae isolated from ready-to-eat shrimps were also found to express hemolytic, lipolytic, DNA degrading activity and also gelatinase production [41]. Additional studies are necessary in order to understand the extent of the possible role of these phenotypes both in animal and in human *Salmonella* infections. Actually, from the obtained data, the possibility of the same bacterial clone infecting different animals and adapting/evolving within the hosts cannot be excluded. Although a molecular based approach would bring valuable information regarding the identity and the possible genetic relationship between the studied isolates, the present report was designed to clarify the therapeutic potential of chlorhexidine, testing one isolate from each animal. Despite the possible genetic similarities, the foremost important assessed feature of each *Salmonella* isolate was the phenotypical behavior, namely the susceptibility to a commonly used biocide, chlorhexidine gluconate. Thus, the information resulting from this study can be adapted and applied in reptile medicine.

In the present study, the occurrence of both bactericidal and bacteriostatic effects of chlorhexidine gluconate is an example of the duality of the antimicrobial effect that takes place according to the applied concentration. Previous reports revealed chlorhexidine gluconate MIC values ranging from 8 to 16 mg/L when tested towards *Salmonella* Bredeney, Dublin, Gallinarum, Montivideo Virshow and Typhimurium [42]. Another study recorded a range of MIC values for *Salmonella* isolates of animal origin (broilers, cattle and pigs) between 2 and 64 mg/L [43]. More recent studies reported MIC values of 1–8 mg/L in turkey *Salmonella* isolates from commercial processing plants, and MIC values below 4 mg/L to 64 mg/L regarding different *Salmonella* serovars isolated from chicken and in egg production chains [6,44,45]. The overall mean MIC value calculated for the studied *Salmonella* spp. isolates from pet reptiles was 11.90 mg/L, which is coherent with those values. The global mean MBC value is approximately three times the mean MIC. The suggestion that both MIC and MBC values should be included in the monitorization of biocidal susceptibility is consistent with the results obtained in this study considering that both values provide complementary information [46].

Although MIC an MBC values are valuable for evaluating the antimicrobial effect of chlorhexidine gluconate, the previous studies were carried out with planktonic cells. The fact that the *Salmonella* spp. isolates are capable of biofilm formation is worrisome, since *Salmonella* organized in biofilms is less susceptible to disinfectants than planktonic cells, with preliminary studies indicating that disinfectants used at an effective concentration for *Salmonella* biofilm reduction can cause the selection of more virulent cells [47]. The high frequency of the studied reptile *Salmonella* isolates capable of forming biofilms (90.63%) is similar to previously reported data. High frequencies of pellicle formation in the air-liquid interface by *Salmonella* Agona (100%), *Salmonella* Montevideo (100%), and *Salmonella* Senftenberg (88%) were already described [48]. However, in the same study, only 55% of the *Salmonella* Typhimurium isolates tested were biofilm producers [48]. On other studies, the expression of biofilm formation by *Salmonella* Typhimurium isolates varied under the same circumstances, with different strains and morphotypes demonstrating different biofilm capabilities [49,50].

Biofilms are common on liquid-hard surfaces interfaces [51], such as in certain type of reptile cages or in aquariums. In order to simulate a more realistic approach to the effects of chlorhexidine gluconate on *Salmonella* cultures, the antimicrobial action of chlorhexidine gluconate activity was tested on the biofilms formed by the reptile *Salmonella* isolates during a 24 h-period. A chlorhexidine gluconate MBIC value within the concentration limits tested was obtained regarding all the *Salmonella* isolates studied. Regarding the *Salmonella* isolates towards which the MBEC values exceeded 714.29 mg/L, chlorhexidine gluconate was simply not effective in terms of eradicating those biofilms. Overall, chlorhexidine gluconate MBIC and MBEC results show that *Salmonella* biofilms are less susceptible to this biocide, what is consistent with a previous report which stated that three-day old *Salmonella* Typhimurium biofilms were less susceptible to chlorhexidine gluconate when compared to the corresponding planktonic cells [52]. 

## 4. Materials and Methods

### 4.1. Sample Collection and Salmonella spp. Isolation

A total of 78 cloacal swabs were obtained from pet reptiles, specifically 43 Chelonians (commonly referred as turtles), 27 Saurians (commonly named lizards), and eight Ophidians (usually known as snakes). The cloacal swabs were performed using cotton swabs in AMIES transport media (VWR, Amadora, Portugal) during routine health check-ups at the house of the owners or at pet shops, all located in the Lisbon Metropolitan Area, Portugal. All animals were cared for according to the rules given by the current EU (Directive 2010/63/EC) and national (DL 113/2013) legislation and by the competent authority (Direção Geral de Alimentação e Veterinária, DGAV, (www.dgv.min-agricultura.pt/portal/page/portal/DGV, accessed on 20 January 2021) in Portugal. Verbal informed consent was obtained from all the owners. Trained veterinarians performed sample collection of all the samples, following standard routine procedures. After collection, swabs were kept under refrigeration conditions (4 °C) for no longer than 48 h until processing in the Microbiology Laboratory of the Veterinary Medicine Faculty—University of Lisbon for *Salmonella* spp. isolation. 

Briefly, each cloacal swab was homogenized and incubated in 5 mL of buffered peptone water (BPW) (Scharlau, Valencia, Spain) for 18 ± 2 h at 37 °C. After the initial incubation, 1 mL of BPW was then added to 10 mL of Muller-Kaufmann Tetrathionate (MKTT) Broth (Oxoid, Hampshire, UK) and incubated for 18–24 h at 37 °C. Simultaneously, 0.1 mL of the BPW solution was added to 10 mL Rappaport–Vassiliadis broth (Oxoid, Hampshire, England) and the resulting suspension was incubated for 18–24 h at 41.5 °C. Afterwards, suspensions were inoculated in Hektoen Agar (Liofilchem, Teramo, Italy) and xylose lysine deoxicholate agar (Scharlau, Valencia, Spain) plates, by streaking, and incubated at 37 °C for 20 ± 2 h. The resulting presumptive *Salmonella* spp. colonies were selected and transferred to triple sugar iron (TSI) Agar (Scharlau, Valencia, Spain) and to urea broth (Oxoid, Dadirlly, France) and incubated for 20 ± 2 h at 37 °C. Presumptive *Salmonella* spp. isolates were identified through the growth pattern in TSI agar and in Urea Broth. The method described is an adaptation of a previously described method [53]. *Salmonella* spp. isolates were identified using biochemical profile system API 20E (BioMérieux, Craponne, France). The biochemical identification was later confirmed by agglutination with Antiserum *Salmonella* OMNIVALENT Omni-O (Bio-Rad Laboratories, Inc., Marnes-la-Coquette, France). 

### 4.2. Antimicrobial Susceptibility Testing

Antimicrobial susceptibility testing was performed by the disk diffusion method, according to Clinical and Laboratory Standards Institute guidelines (CLSI) [54]. The tested antibiotics were amoxicillin/clavulanic acid (AMC, 30 μg), ampicillin (AMP, 10 μg), amikacin (AK, 30 μg), chloramphenicol (C, 30 μg), gentamicin (CN, 10 μg), cefotaxime (CTX, 30 μg), enrofloxacin (ENR, 5 μg), nalidixic acid (NA, 30 μg), penicillin (P, 10 U), ciprofloxacin (CIP, 5 μg), sulfamethoxazole/trimethoprim (SXT, 25 μg), and tetracycline (TE, 30 μg). All antibiotics were purchased from Oxoid, Dadirlly, France. *Escherichia coli* ATCC 25922 was used as the control strain for test performance. Multidrug resistance (MDR) phenotype was considered to be present whenever an isolate revealed resistance to three or more antimicrobial compounds belonging to different classes [55].

### 4.3. Virulence Phenotype Analysis

In order to assess the virulence phenotype of the *Salmonella* isolates, plate tests were performed for evaluating their DNase, gelatinase, hemolytic and lipase activities.

DNase activity testing was performed by streaking the bacterial isolates on DNase test Agar plates (Liofilchem, Teramo, Italy) supplemented with 0.01% toluidine blue. The plates were incubated for 48 h at 37 °C and positive results showed a transparent halo surrounding the colonies.

Gelatinase activity was tested by streaking the isolates on Gelatinase test Agar plates (Liofilchem, Teramo, Italy), followed by incubation at 37 °C for 48 h. Afterwards, plates were flooded with a mercury chloride solution and the gelatinase positive isolates showed a transparent halo around the colonies.

Production of hemolysins was determined by streaking the isolates on Columbia Agar plates supplemented with 5% sheep blood (BioMérieux, Craponne, France) and incubated for 48 h at 37 °C. The presence of clear halos surrounding the colonies was interpreted as β-hemolysis.

Lipase activity testing was achieved by culturing the isolates in Spirit Blue Agar plates (Difco, Algés, Portugal) supplemented with Tween 80 (30 g/L) and incubating for 48 h at 37 °C. Lipase producing isolates exhibited clear halos around the colonies.

### 4.4. Chlorhexidine Gluconate Minimum Inhibitory Concentration and Minimum Bactericidal Concentration Determination

The in vitro susceptibility profile of the *Salmonella* isolates to chlorhexidine gluconate was assessed by an adapted protocol based on the microtiter broth dilution method [56,57]. Isolates were grown in a nonselective brain heart infusion (BHI) agar medium (VWR Chemicals, Leuven, Belgium) at 37 °C for 24 h. Bacterial suspensions with 10^8^ CFU/mL were prepared directly from plate cultures in sterile normal saline (Merck, Germany) to a 0.5 McFarland suspension. The bacterial suspensions were then diluted in fresh BHI broth (VWR Chemicals, Leuven, Belgium) to a concentration of 10^7^ CFU/mL. 

Chlorhexidine gluconate dilutions were prepared from a stock solution at a concentration of 4% (*w*/*v*) (AGA, Lisboa, Portugal). A volume of 25 μL of chlorhexidine gluconate at 0.5, 0.1, 0.05, 0.01, 0.005 and 0.001% were distributed in 96-well flat-bottomed polystyrene microtiter plates (Nunc, Thermo Fisher Scientific, Roskilde, Denmark), apart from the negative and positive controls. All the wells were inoculated with 150 μL of the 10^7^ CFU/mL bacterial suspensions, with exception of the negative control wells, which contained only broth medium. Therefore, the final concentration of chlorhexidine gluconate in the wells corresponded to 714.28, 142.86, 71.43, 14.29, 7.14, and 1.43 mg/L. Afterwards, microplates were statically incubated for 24 h at 37 °C. The minimum inhibitory concentration (MIC) was determined as the lowest concentration of chlorhexidine gluconate that visually inhibited microbial growth. 

The minimum bactericidal concentration (MBC) value was assessed by inoculating 3 μL of the suspensions from the wells were no growth was observed on BHI agar plates, which were incubated at 37 °C for 24 h. MBC was determined as the lowest chlorhexidine gluconate concentration that did not allow colony development [57,58].

The ratio between MBC and MIC was calculated in order to determine the antimicrobial effect of chlorhexidine gluconate. The effect was considered to be bactericidal when the MBC was no more than fourfold the MIC, or bacteriostatic when the ratio exceeded four [58].

All experiments were conducted in duplicate and independent assays were performed at least three times in different dates.

### 4.5. Biofilm Formation in the Air-Liquid Interface

Biofilm forming ability was assessed through a biofilm formation assay in the air–liquid interface, by inoculating 0.5 mL of an overnight BHI broth culture, adjusted to a 0.5 McFarland standard, in a 4.5 mL of Luria broth (LB) without NaCl (1:10), prepared using yeast extract (Oxoid, Hampshire, England) and bacto tryptone (BD, Oeiras, Portugal). Isolates were incubated at 28 °C for eight days and each isolate was visually examined for pellicle formation on a daily basis [49]. The isolates capable of forming a pellicle in two distinct occasions were considered to be positive for biofilm formation, and the number of days required until the pellicle was perceivable was used to calculate the mean time for biofilm formation.

All assays were repeated in three independent dates, including 10% replicates.

### 4.6. Chlorhexidine Gluconate Minimum Biofilm Inhibitory Concentration and Minimum Biofilm Eradication Concentration Determination

The antimicrobial susceptibility of the *Salmonella* isolates when embedded in a 24 h biofilm was evaluated by a modified version of the Calgary Biofilm Pin Lid Device [57,59]. For minimum biofilm inhibitory concentration (MBIC) and minimum biofilm eradication concentration (MBEC) assays, the bacterial isolates were grown in BHI agar medium (VWR Chemicals, Leuven, Belgium) at 37 °C for 24 h. Bacterial suspensions with approximately 10^8^ CFU/mL were prepared directly from plate cultures in sterile normal saline (Merck, Darmstadt, Germany) by comparison with a 0.5 McFarland standard (BioMérieux, Craponne, France). Suspensions were then diluted in fresh BHI broth (VWR Chemicals, Leuven, Belgium) to a concentration of 10^6^ CFU/mL. Then, 175 μL of the bacterial suspensions were distributed in 96-well flat-bottomed polystyrene microtiter plates, covered with 96-peg polystyrene lids (Nunc-TSP; Thermo Fisher Scientific, Roskilde, Denmark) and statically incubated for 24 h at 37 °C, allowing biofilm formation on the pegs. Peg lids were then rinsed three times in sterile normal saline to remove planktonic bacteria and placed on new microplates containing the set of chlorhexidine gluconate solutions previously described, corresponding to a final concentration by well of 714.28, 142.86, 71.43, 14.29, 7.14, and 1.43 mg/L. 

Microplates were again incubated for 24 h at 37 °C, without shaking. After incubation, peg lids were removed, and the MBIC value was determined as the lowest chlorhexidine gluconate concentration that visually inhibited microbial growth. Subsequently, in order to determine the MBEC value, peg lids were rinsed three times in sterile normal saline, placed in new microplates containing only 175 μL of fresh BHI medium and incubated in an ultrasound bath (Grant MXB14, Essex, England), at 50 Hz during 15 min in order to disperse the biofilm-based bacteria from the peg surface. Afterwards, peg lids were discarded, and microplates were covered with normal lids and incubated for 24 h at 37 °C. The MBEC value was determined through direct observation of bacterial growth in the wells and defined as the lowest chlorhexidine gluconate concentration that visually eliminates the microbial growth [57]. 

Experiments were conducted in duplicate and independent assays were performed at least two times on different dates.

### 4.7. Statistical Analysis

For statistical analysis, the associations between frequency of *Salmonella* isolation and reptile group, AMR *Salmonella* and reptile group and virulence phenotype and reptile group were evaluated using the Fisher exact test. Association between different MIC, MEC and MBIC values of chlorhexidine gluconate on *Salmonella* isolates, the number of days until biofilm formation and the reptiles group was assessed recurring to the Brown–Forsythe robustness test based on a one-way ANOVA test. All statistical tests were performed on IBM SPSS Statistical program version 26 for Windows (SPSS Inc., Chicago, IL, USA). Associations were considered to be significant whenever *P* values were less than 0.05. 

## 5. Conclusions

The present study reports the isolation of *Salmonella* from healthy pet reptiles and stresses their possible role in human non-typhoidal salmonellosis cases. Although presenting high levels of antimicrobial susceptibility, the expression of phenotypical virulence traits and the ability to form biofilms by these isolates are worrisome. Pet reptile owners should always employ good hygiene practices whenever manipulating the animals, but also when in contact with the environment in which the animals are kept. Overall, the use of chlorhexidine gluconate was considered to be effective, both in planktonic cells and biofilms, pointing out the potential of this biocide’s use in reptile clinics.

## Figures and Tables

**Table 1 antibiotics-10-00324-t001:** *Salmonella* positive animals, divided by category and species.

Category	Species	Number of Positive Animals
Ophidians	*Pantherophis guttatus guttatus*	2
	*Python regius*	2
Chelonians	*Centrochelys sulcata*	1
	*Chelonoidis carbonaria*	1
	*Geochelone sulcata*	1
	*Pseudemys spp.*	2
	*Sternotherus odoratus*	1
	*Testudo horsfield*	1
	*Traquemys scripta elegans*	2
Saurians	*Chlamydosaurus kingii*	2
	*Ctenosaura quinquecarinata*	1
	*Gerrhosaurus major*	1
	*Hydrosaurus amboinensis*	1
	*Iguana iguana*	1
	*Physignatus cocincinus*	3
	*Physignatus lesueurii lesueurii*	1
	*Pogona vitticeps*	8
	*Tupinambis rufrescens*	1

**Table 2 antibiotics-10-00324-t002:** Detailed information regarding the *Salmonella* isolates under study.

Isolate Number	Group	Species	Owner	API20E Result
4	Ophidian	*Python regius*	A	*Salmonella enterica* subsp. *arizonae*
12	Chelonian	*Pseudemys* spp.	B	*Salmonella* spp.
21	Ophidian	*Pantherophis guttatus guttatus*	C	*Salmonella* spp.
26	Chelonian	*Geochelone sulcata*	D	*Salmonella enterica* subsp. *arizonae*
27	Chelonian	*Chelonoidis carbonaria*	D	*Salmonella enterica* subsp. *arizonae*
30	Saurian	*Pogona vitticeps*	E	*Salmonella* spp.
31	Saurian	*Pogona vitticeps*	E	*Salmonella* spp.
32	Saurian	*Pogona vitticeps*	E	*Salmonella* spp.
33	Saurian	*Physignatus cocincinus*	E	*Salmonella* spp.
34	Saurian	*Pogona vitticeps*	E	*Salmonella* spp.
35	Chelonian	*Centrochelys sulcata*	F	*Salmonella* spp.
36	Chelonian	*Testudo horsfield*	F	*Salmonella* spp.
41	Chelonian	*Sternotherus odoratus*	F	*Salmonella* spp.
44	Chelonian	*Pseudemys* spp.	G	*Salmonella* spp.
46	Saurian	*Pogona vitticeps*	H	*Salmonella* spp.
47	Chelonian	*Traquemys scripta elegans*	I	*Salmonella enterica* subsp. *arizonae*
48	Chelonian	*Traquemys scripta elegans*	I	*Salmonella* spp.
50	Saurian	*Ctenosaura quinquecarinata*	J	*Salmonella enterica* subsp. *arizonae*
52	Saurian	*Physignatus cocincinus*	J	*Salmonella enterica* subsp. *arizonae*
53	Saurian	*Physignatus cocincinus*	J	*Salmonella einterica* subsp. *arizonae*
54	Saurian	*Tupinambis rufrescens*	J	*Salmonella enterica* subsp. *arizonae*
55	Saurian	*Pogona vitticeps*	J	*Salmonella* spp.
56	Saurian	*Pogona vitticeps*	J	*Salmonella enterica* subsp. *arizonae*
58	Saurian	*Gerrhosaurus major*	J	*Salmonella enterica* subsp. *arizonae*
61	Saurian	*Hydrosaurus amboinensis*	J	*Salmonella enterica* subsp. *arizonae*
62	Saurian	*Chlamydosaurus kingii*	J	*Salmonella enterica* subsp. *arizonae*
63	Saurian	*Chlamydosaurus kingii*	J	*Salmonella* spp.
66	Saurian	*Physignatus lesueurii lesueurii*	J	*Salmonella enterica* subsp. *arizonae*
69	Saurian	*Iguana iguana*	J	*Salmonella enterica* subsp. *arizonae*
70	Ophidian	*Pyton regius*	K	*Salmonella enterica* subsp. *arizonae*
73	Ophidian	*Pantherophis guttatus guttatus*	K	*Salmonella* spp.
76	Saurian	*Pogona vitticeps*	L	*Salmonella* spp.

**Table 3 antibiotics-10-00324-t003:** Antimicrobial resistance and virulence phenotype results.

Antimicrobial Resistance	Ophidians (%)	Chelonians (%)	Saurians (%)	*p* Value
**AMC**	0 (0%)	3 (33.3%)	0 (0%)	0.0286
**AMP**	0 (0%)	3 (33.3%)	0 (0%)	0.0286
**AK**	0 (0%)	1 (11.1%)	0 (0%)	N.S.
**C**	0 (0%)	0 (0%)	1 (5.26%)	N.S.
**CN**	0 (0%)	0 (0%)	0 (0%)	-
**CTX**	0 (0%)	0 (0%)	0 (0%)	-
**ENR**	0 (0%)	0 (0%)	0 (0%)	-
**NA**	0 (0%)	1 (11.1%)	1 (5.26%)	N.S.
**P**	4 (100%)	8 (88.89%)	19 (100%)	N.S.
**CIP**	0 (0%)	0 (0%)	0 (0%)	-
**SXT**	0 (0%)	0 (0%)	1 (5.26%)	N.S.
**TE**	0 (0%)	1 (11.1%)	0 (0%)	N.S.
**Virulence phenotype**				
**Hemolytic activity**	4 (100%)	9 (100%)	19 (100%)	-
**Lipolytic activity**	4 (100%)	9 (100%)	19 (100%)	-
**DNase activity**	4 (100%)	4 (44.44%)	11 (57.89%)	N.S.
**Gelatinolytic activity**	0 (0%)	0 (0%)	0 (0%)	-

Abbreviations: AMC, amoxicillin/clavulanic acid; AMP, ampicillin; AK, amikacin; C, chloramphenicol; CN, gentamicin; CTX, cefotaxime; ENR, enrofloxacin; NA, nalidixic acid; P, penicillin; CIP, ciprofloxacin; SXT, sulfamethoxazole/trimethoprim; TE, tetracycline; N.S., non-significant.

**Table 4 antibiotics-10-00324-t004:** Chlorhexidine gluconate minimum inhibitory concentrations, minimum bactericidal concentrations, minimum biofilm inhibitory concentrations, minimum biofilm eradication concentrations and biofilm formation results.

Heading	Ophidians	Chelonians	Saurians	*p* Value
**MIC (mg/L)**	11.98 ± 1.46	11.25 ± 4.66	12.19 ± 3.44	N.S.
**MBC (mg/L)**	86.84 ± 72.75	27.87 ± 11.71	33.87 ± 52.91	N.S.
**MBIC (mg/L)**	57.15 ± 28.57	64.02 ± 12.32	72.87 ±39.60	N.S.
**MBEC (mg/L)**	244.05 ± 131.49 *	333.65 ± 222.2 *	397.39 ± 194.74 *	N.S.
**Biofilm formation (days)**	5.1 ± 0.49	4.7 ± 1.0	4.2 ± 0.79	N.S.

Abbreviations: MIC, minimum inhibitory concentration; MBC, minimum bactericidal concentration; MBIC, minimum biofilm inhibitory concentration; MBEC, minimum biofilm eradication concentration; N.S., non-significant. * Values above 714.29 mg/L were not included.

## Data Availability

The data presented in this study are available in Appendix A.

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
