# Peer review of "Salmonella spp. in Pet Reptiles in Portugal: Prevalence and Chlorhexidine Gluconate Antimicrobial Efficacy"

_antibiotics, 2021, doi:10.3390/antibiotics10030324_

Round 1
Reviewer 1 Report
The aim of the research was to assesses the antimicrobial susceptibility profiles of some Salmonella isolates collected from pet animal, and the production of microbial factors with potential role in virulence . Additionally, the authors evaluated the antimicrobial effects of the chlorhexidine against the bacterial isolates of Salmonella. Overall the methods of research were appropriate, there are reference to earlier, similar studies from different regions, different industries or different disciplines. The antimicrobial susceptibility testing results revealed low antibiotic resistance rates among the analysed strains. Table 3 is a bit confusing for me, i.e the title refers to resistance, yet in the first column the term susceptibility it is used. Also, authors should distinguish between intrinsic antibiotic resistance and acquired resistance. The effects of chlorhexidine against Salmonella are well described in the literature, hence the originality is poor.
Author Response
Table 3 is a bit confusing for me, i.e the title refers to resistance, yet in the first column the term susceptibility it is used.
The term susceptibility was removed and the term resistance was inserted.
Also, authors should distinguish between intrinsic antibiotic resistance and acquired resistance.
The difference between intrinsic and acquired antibiotic resistance was addressed on the Introduction section.
The effects of chlorhexidine against Salmonella are well described in the literature, hence the originality is poor.
Indeed the effects of chlorhexidine on Salmonella isolates of multiple origins have been described in the literature, yet that is not the case regarding reptile-associated isolates. Furthermore, the assessment of the effective concentrations of chlorhexidine on reptile-associated Salmonella isolates is very important in order to establish therapeutic protocols adjusted to veterinary medicine, and more specifically in reptile medicine.
Reviewer 2 Report
In the manuscript antibiotics-1145620 by Cota and colleagues, the authors have investigated the occurrence of Salmonella spp. in the intestinal microbiota of reptiles, hold as pets, the incidence of antimicrobial resistant strains and the antimicrobial efficacy of chlorhexidine gluconate, usually adopted as disinfectant in both human and vet therapy. Although different Salmonella strains were isolated, no specific correlation between reptile species and bacterial carriage was observed; the recovery of antibiotic resistant isolates, mostly to β-lactams, was low and no strain was multidrug resistant, although almost all the strains exhibited virulent traits, specifically haemolytic and lipolytic activities. Overall, chlorhexidine gluconate showed a good antimicrobial activity, against both planktonic and biofilm-embedded Salmonella cells, despite a high variability of the results.
The work is interesting and sounding, especially considering the increasing trend in keeping reptiles as pets and the One Health vision of antimicrobial resistance spread. The authors have used and described in detail a variety of microbiological techniques, the results are clearly presented, well supported by statistical analysis and well commented in the discussion session. Nevertheless, there are some issue that deserve to be better explained or revised:
-The bacterial species identification was performed by biochemical assays and confirmed by immunological tests. The authors should consider confirming their results by molecular identification, i.e. sequencing of 16s rDNA, even to improve the detection of different Salmonella species;
-Since, as mentioned by the authors in the Results section and in Table 2 and S1, different Salmonella strains were recovered from different animals of the same owners, the isolates belonging to the same species, or not identified at species level, should be typed. Indeed, from the observed data, the presence of the same bacterial clone infecting different animals and adapting/evolving within the hosts cannot be excluded.
-For chlorhexidine gluconate resistance determination, the reported values showed a wide range of variability; the authors should consider, where possible, to indicate the minimal disinfectant concentrations working on the 50% or 90% of the tested bacterial strains rather than the average values.
For these reasons, major revisions are recommended for the paper before being published in “Antibiotics”.
MINOR COMMENTS
Line 11, please delete the term “ prevalence” after “Salmonella spp.”;
Line 35, please delete the term “and” in “with and their estimated numbers” and “to” before “up to”;
Line 37, please correct “on the environments” with “in the environments”;
Line 41, please correct “from the more than 2500 known Salmonella serotypes” with “among more than 2500 known Salmonella serotypes”;
Line 53, please define well the abbreviation AMR: if it is for “antimicrobial resistance”, as reported above, please correct with “antimicrobial resistant”;
Line 148, what do the authors mean by “were considered to be statistically different (p = 0.257)”? Please clarify the sentence.
Line 154, please delete “sp” in “Salmonella sp. isolates”;
Line 204, please correct “Salmonella isolates from reptiles are known to be resistance” with “Salmonella isolates from reptiles are known to be resistant”;
Line 209, please delete the term “of” in “All of the isolates” and “a” before “hemolytic and lipolytic behaviors“;
Line 213, please correct “in Salmonella enterica serovar Typhimurium”;
Lines 221, 222, please correct “did not express”;
Line 228, please correct “to the applied concentration”;
Line 244, please correct “Salmonella organized in biofilms are less susceptible” with “Salmonella organized in biofilms is less susceptible”;
Lines 277, 286 and 288, please correct “Salmonella sp.” With “Salmonella spp.”;
Line 280, please correct “1 ml of BPW was then added”;
Line 300, please correct “tetracyclin” with “tetracycline”;
Line 301, please correct “Oxoid, France. Escherichia coli ATCC 25922”;
Line 400, please type Salmonella.
Please provide a better quality table for Table S1, including a title and explanation for all the abbreviations.
Author Response
-The bacterial species identification was performed by biochemical assays and confirmed by immunological tests. The authors should consider confirming their results by molecular identification, i.e. sequencing of 16s rDNA, even to improve the detection of different Salmonella species;
Although we agree that a molecular identification would provide further information regarding the tested isolates, unfortunately at the moment it is not feasible to implement other methods as suggested, since the project and the associated funding have ended, nor the time-frame imposed by the Journal to address the reviewers questions allows such studies.
-Since, as mentioned by the authors in the Results section and in Table 2 and S1, different Salmonella strains were recovered from different animals of the same owners, the isolates belonging to the same species, or not identified at species level, should be typed. Indeed, from the observed data, the presence of the same bacterial clone infecting different animals and adapting/evolving within the hosts cannot be excluded.
We agree, and a simmilar statement was added to the discussion of the results.
-For chlorhexidine gluconate resistance determination, the reported values showed a wide range of variability; the authors should consider, where possible, to indicate the minimal disinfectant concentrations working on the 50% or 90% of the tested bacterial strains rather than the average values.
The overall median values were included in the revised version of the Manuscript.
MINOR COMMENTS
All of the issues pointed out were corrected.
-Please provide a better quality table for Table S1, including a title and explanation for all the abbreviations.
Table S1 was revised accordingly.
Reviewer 3 Report
The manuscript submitted for review addresses an interesting and often overlooked problem of the spread of salmonellosis via reptiles.
The manuscript requires a thorough linguistic check and proofreading by a native speaker.
The manuscript is more of a simple report than a research paper. The assessment of genetic similarity, which is necessary in this type of work, is very lacking, carried out on the basis of the PFGE technique or at least RAPD. There are also no genetic analysis related to, for example, genes encoding drug resistance or virulence factors, which is a standard nowadays. I strongly encourage the Authors to complete this type of research.
Detailed recommendations are provided below:
INTRODUCTION:
- Provide information on how many cases of salmonellosis are currently recorded in the world and in Europe and what is the trend of the disease in recent years.
- How many cases of salmonellosis caused by reptilian strains are there? Are there any known epidemics related to this source?
MATERIALS AND METHODS:
- Do the Authors have the consent of the appropriate bioethic committee? Provide the document number
- From how many different keepers swabs were obtained? How many and what animals were there at the particular owner? Please specify in the text.
- What swabs and from what material were used in the research. Please specify in the text.
- Is the presented method of dealing with swabs compliant with any standard or is it an own method?
- Why were the CLSI recommendations used and not EUCAST?
- In the title of point 4.4. it should be stated that it is about chlorhexidine
- Please describe the procedure for determining the MBC more detailed
RESULTS:
- Table 2 - The ISOLATES heading can be confusing and indicate that this is about the amount of isolates and not their sequential numbers.
- Line 135 – it should be: „the differences were not statistically significant”
Author Response
The assessment of genetic similarity, which is necessary in this type of work, is very lacking, carried out on the basis of the PFGE technique or at least RAPD. There are also no genetic analysis related to, for example, genes encoding drug resistance or virulence factors, which is a standard nowadays. I strongly encourage the Authors to complete this type of research.
Although we agree that analysis by other methods such as PFGE or RAPD would provide further information regarding the tested isolates, as well as other molecular analysis, at the moment it is not feasible to implement those methods as suggested, since the project and associated funding have ended, nor the time-frame imposed by the Journal to address the reviewers questions allows such studies.
INTRODUCTION:
Provide information on how many cases of salmonellosis are currently recorded in the world and in Europe and what is the trend of the disease in recent years.
Such information was added to the revised version of the manuscript.
How many cases of salmonellosis caused by reptilian strains are there? Are there any known epidemics related to this source?
Such information was added to the revised version of the manuscript.
MATERIALS AND METHODS:
Do the Authors have the consent of the appropriate bioethic committee? Provide the document number
All animals were cared for according to the rules given by the current EU (Directive 2010/63/EC) and national (DL 113/2013) legislation and by the competent authority (Direção Geral de Alimentação e Veterinária, DGAV, (www.dgv.min-agricultura.pt/portal/page/portal/DGV) in Portugal. Only noninvasive samples were collected during routine procedures with consent of owners, and no ethics committee approval was needed. Trained veterinarians obtained all the samples, following standard routine procedures. No animal experiment has been performed in the scope of this research. Verbal informed consent was obtained from all the owners. As some of the participants in the study were unaccustomed to deal with forms, all the necessary information about the study was provided to all the participants before obtaining their consent.
From how many different keepers swabs were obtained? How many and what animals were there at the particular owner? Please specify in the text.
The disclosure of information regarding Salmonella negative animals was not discussed with the participants of the study, thus no consent regarding that matter was given and therefore that data was not included. Nevertheless, other information, such as the different cities of the Lisbon Metropolitan area where the animals live can be provided is necessary.
What swabs and from what material were used in the research. Please specify in the text.
Cotton swabs with AMIES media. The information was added to the text.
Is the presented method of dealing with swabs compliant with any standard or is it an own method?
The method described is an adaptation of a previously described method (Hendriksen, R.S. (2003). Laboratory protocols- level 1 Training Course Isolation of Salmonella. Global Salm- Surv, 4th Ed.)
Why were the CLSI recommendations used and not EUCAST?
The CLSI recommendations were used since EUCAST does not provide clearly defined limits for isolates of veterinary origin.
In the title of point 4.4. it should be stated that it is about chlorhexidine
Corrected in the revised manuscript.
Please describe the procedure for determining the MBC more detailed
As mentioned in the text, the Minimum Bactericidal Concentration (MBC) value was assessed by inoculating 3 μL of the suspensions from the wells were no growth was observed on BHI agar plates, which were incubated at 37°C for 24 h. MBC was determined as the lowest chlorhexidine gluconate concentration that did not allow colony development.
RESULTS:
Table 2 - The ISOLATES heading can be confusing and indicate that this is about the amount of isolates and not their sequential numbers.
The referred heading was changed in the revised version to Isolate number.
Line 135 – it should be: „the differences were not statistically significant”
Corrected in the revised manuscript.
Round 2
Reviewer 2 Report
In the revised version of the manuscript antibiotics-1145620 by Cota and colleagues, the authors have tried to address the raised issues. Unfortunately, the correct bacterial identification by molecular methods and the strain typing are pivotal to efficiently analyse the obtained data. A special focus must be dedicated to the strain typing: how can the authors exclude that they have recovered the same few strains from different animals belonging to the same owner, in particular considering the results reported in lines 89-93 of the revised manuscript? This could even affect the statistical significance of the presented results.
Thus, although the difficulties reported by the authors are acknowledged, these two revisions are still recommended for the manuscript publication, or the rationale for not performing the related experiments must be detailed in the discussion section.
Author Response
In the revised version of the manuscript antibiotics-1145620 by Cota and colleagues, the authors have tried to address the raised issues. Unfortunately, the correct bacterial identification by molecular methods and the strain typing are pivotal to efficiently analyse the obtained data. A special focus must be dedicated to the strain typing: how can the authors exclude that they have recovered the same few strains from different animals belonging to the same owner, in particular considering the results reported in lines 89-93 of the revised manuscript? This could even affect the statistical significance of the presented results.
Thus, although the difficulties reported by the authors are acknowledged, these two revisions are still recommended for the manuscript publication, or the rationale for not performing the related experiments must be detailed in the discussion section.
Regarding the above mentioned comments/suggestions, the following sentences were added to the discussion section (lines 237-243):
"Although a molecular based approach would bring valuable information regarding the identity and the possible genetic relationship between the studied isolates, the present report was designed to clarify the therapeutic potential of chlorhexidine, testing one isolate from each animal. Despite the possible genetic similarities, the foremost important assessed feature of each Salmonella isolate was the phenotypical behavior, namely the susceptibility to a commonly used biocide, chlorhexidine gluconate. Thus, the information resulting from this study can be adapted and applied in reptile medicine."
Reviewer 3 Report
Please add this information in manuscript:
The method described is an adaptation of a previously described method (Hendriksen, R.S. (2003). Laboratory protocols - level 1 Training Course Isolation of Salmonella. Global Salm- Surv, 4th Ed.)
Author Response
Please add this information in manuscript:
The method described is an adaptation of a previously described method (Hendriksen, R.S. (2003). Laboratory protocols - level 1 Training Course Isolation of Salmonella. Global Salm- Surv, 4th Ed.)
The requested information was added to the manuscript (Reference 54).